# Chemistry and Reactivity of Tannins in Vitis spp.: A Review

**DOI:** 10.3390/molecules25092110

**Published:** 2020-04-30

**Authors:** Aude A. Watrelot, Erin L. Norton

**Affiliations:** 1Department of Food Science and Human Nutrition, Iowa State University, 536 Farm House Lane, Ames, IA 50011-1054, USA; elnorton@iastate.edu; 2Midwest Grape and Wine Industry Institute, Iowa State University, 536 Farm House Lane, Ames, IA 50011-1054, USA

**Keywords:** *Vitis vinifera*, *Vitis labrusca*, interspecific hybrid grapes, polyphenols, reactivity, proteins, cell wall material

## Abstract

Tannins are a group of polyphenols found in fruits, leaves, trees, etc., well known in the leather industry and in apples, persimmons and grapes, because of their capacity to interact with other polyphenols or other components either from the food product or from saliva. Prior to being able to interact with other compounds, tannins have to be extracted from the food matrix, which depends on their chemistry, as well as the chemical structure of other components, such as cell wall material and proteins. *Vitis vinifera* grapes are commonly grown around the world and are used in winemaking, providing good quality wines with different levels of tannins responsible for the final wine’s astringency. Many studies have focused on tannins extractability and retention with cell wall material, and the reactivity of tannins with proteins in *Vitis vinifera* grapes and wine, but there are very few reports for other Vitis species. However, depending on the environmental characteristics of certain regions, Vitis hybrid grapes are grown and used to produce wines more and more. This review focuses on the comparison of the chemistry of tannins, and their reactivity with other macromolecules in Vitis species.

## 1. Introduction

Tannins are plant polyphenolic secondary metabolites used during Classical Antiquity in the treatment of animal skins to avoid their putrefaction. Due to the interaction between tannins and the collagen from the skins, the collagen was stabilized and the animal skins were tanned and transformed into leather [1]. Since this time, the tanning process, which is the production of leather from the vegetal source of tannins, continues to be used. Therefore, tannins are molecules able to interact and precipitate proteins, among other molecules, such as polysaccharides and polyphenols [2]. Hydrolyzable tannins, condensed tannins and phlorotannins are the main groups of tannins that are found in various parts of the higher plants. As natural products in plants, they act as a natural barrier against insects, pathogens and animals, because of their ability to react with proteins and their antioxidant properties [3]. Therefore, tannins contribute to the reduction of cardiovascular diseases and of some cancer risks.

Depending on their chemical structure, they can be found in specific botanical sources and/or plant parts. Hydrolyzable tannins are commonly found in wood or stems, while condensed tannins are mostly found in fruits and leaves, such as grapes and apples or tea leaves. In grapes, hydrolyzable tannins can be found in the stem and in the skin [4], while condensed tannins are in the stem, skin, flesh and seeds of the berries [5]. The common grapes grown in the world and used for wine are *Vitis vinifera* grapes. Following the introduction of the North American pest phylloxera and powdery mildew that devastated the Europe grape industry in 1800s, new resistant hybrid varieties were developed by crossing French *Vitis vinifera* with wild American *Vitis.* The most common wild American varieties used for breeding were *Vitis riparia, Vitis rupestris, Vitis berlandieri* and *Vitis labrusca*, but the latter provides “foxy-smelling” aromas to the wine produced [6]. Today, these interspecific hybrid grapes are commonly used in cold regions, because of their resistance to the extreme climate of harsh cold winters and hot humid summers. Therefore, the prevalence of these varieties is growing in the world, especially in cold-hardy regions such as in the U.S. Midwest region. The phenolic composition in grapes and wine is very important, as it allows the determination of the wine color, the evolution of wine and the final wine perception by the consumers. Wine produced from cold hardy red grape varieties are often perceived as acidic, due to the high malic acid content [7] and without mouthfeel or texture, due to the low concentration of tannins in the wine, as well as the different chemical structure of these tannins in the cold-hardy red grape varieties. The extraction and/or the retention of tannins from those grapes to the cell wall material, including proteins and polysaccharides, is currently under investigation and compared with tannins from the *Vitis vinifera* grape varieties, for which much more information is provided.

The goal of this review is to provide an overview of the differences between *Vitis vinifera* and interspecific hybrid grapes and wine, as well as the chemistry of their tannins. The reactivity of tannins with proteins and polysaccharides is detailed to provide potential explanations of the limited tannin extraction in the hybrid cultivar wines.

## 2. Grape Species

A good understanding of the composition of grapes is needed when discussing tannin extractability and retention and the final wine quality. In the following section, the berry cell wall structure and composition in *Vitis vinifera* and Vitis spp., as well as the chemical composition of those berries, including organic acids, sugars and anthocyanins, will be discussed.

The basic physical structure of a grape berry is consistent across all species and is comprised of the skin, pulp, and seeds [8]. The outermost layer of the berry (i.e., the exocarp) is ten to twelve cellular layers thick and covered by a waterproof, waxy cuticle layer. The pulp (i.e., the mesocarp) is comprised of three times as many cellular layers compared to the skin, and the cells are twice as large by the end of the berry growth stage. Vacuoles contain juice account for ninety percent of a mesocarp cell. Cell walls of the mesocarp are also thinner than cells found in the berry skin, and undergo structural changes responsible for berry softening from véraison onwards. Within the mesocarp is the innermost part of the berry, which contains two to four seeds (i.e., the endocarp). A fine layer of cells encases the seeds. Vascular tissues are located centrally with the function of transportation of nutrients to the seeds and are bundled like chicken wire just beneath the exocarp to transport sugar from véraison onwards [9]. During berry development and ripening, the cell wall network composed of hemicelluloses, pectins, cellulose microfibrils and structural proteins undergoes chemical breakdown. Some acidification and enzymatic hydrolysis of these polysaccharides leads to their solubilization and wall loosening. This is also accompanied by the formation of phenolic cross-linking between pectins and hemicelluloses catalyzed by a peroxidase [10,11]. Much more research has been focused on the modification of cell walls in the mesocarp of grape berries, rather than in the skin or exocarp and the seeds or endocarp. From a general standpoint, the berry ripening tends to not change the cell wall thickness of mesocarp cells in *Vitis vinifera* grapes, but leads to an increase in protein content, especially hydroxyproline after véraison, an increase in galacturonan content, the backbone of pectins, becoming more soluble and a decrease in cellulose content [12,13]. Similar observations were also made in Golden Muscat grape skins (*Vitis vinifera × Vitis labrusca*) [11]. The skin and pulp cell walls have a different structure and composition between *Vitis vinifera* grape varieties. Monastrell berries have firmer pulp and skin related to a higher cell wall material and neutral sugars content, compared to Cabernet sauvignon and Syrah berries [14,15]. Cabernet sauvignon, Merlot and Syrah berries have thinner skin cell walls, but Syrah berries showed the lowest degree of acetylation of pectins. It has also been observed by the same authors that the level of enzymes acting on the cell wall degradation were not the same in the grape varieties, i.e., the activity of two galactosidases in Cabernet sauvignon were higher than in Monastrell grapes [12,14,15]. It has been concluded that the structure and composition of cell wall material is different for the *Vitis vinifera* varieties, which might lead to the differences observed of phenolic compounds in the respective wines, as explained later in this review.

When producing wine, the soluble solids, including sugars and organic acids from the pulp or mesocarp, are important parameters used to evaluate the grape maturity. Those chemical characteristics are also associated with pH of grapes which, in turn, has an influence on the phenolic compounds such as anthocyanins, responsible for the red grape color and on microbiological stability. In 1966, 10 sugars and 23 organic acids were identified in Thompson seedless grapes (*V. vinifera*), and their levels were evaluated in different grapevine parts during development and ripening [16]. The main monosaccharides found in grape berries are glucose and fructose. They are consumed by yeast, preferentially glucose, during the alcoholic fermentation process, and the concentration of those sugars is used by winemakers to estimate the alcohol content in the wine. During berry ripening, the sugar content increases rapidly, and the accumulation slows down a week before harvest. The glucose to fructose ratio tends to decrease during ripening, as the level of fructose increases more rapidly than glucose. In a study from the same author, the glucose to fructose ratio was lower in *Vitis labrusca* grapes than in *Vitis vinifera, Vitis riparia*, and *Vitis aestivalis*, ranging from 0.47 to 1.12 [7]. Only in two species, *Vitis champinii* and *Vitis doaniana*, was the glucose content higher than the fructose [7]. In a recent study, the ratio was lower in Brianna grapes that have *Vitis labrusca* in their heritage than in Frontenac grapes (0.88 and 1.10, respectively) [17]. It has been well established that the increase in sugar content during berry ripening is associated with the development of aromatic compounds, as well as with a decrease in acidity. The pH and the titratable acidity are in close relationship with the solubility of tartaric salts in the wine, the color stabilization and the depolymerization and condensation reactions with tannins [18,19]. The main organic acids found in grapes are tartaric, malic and citric acids. Their concentration depends on the grape variety, the harvest date, the climatic conditions, the viticultural practices, etc. [7,16,17], and is highly related to the pH. During grape berry ripening, the concentration of organic acids tends to decrease mostly due to a decrease in the malic acid. In a recently published article, the tartaric acid to malic acid ratio was 0.77, 0.20, 0.38, and 0.56 at harvest in 2016 in Brianna, La Crescent, Frontenac, and Marquette grapes, respectively [17]. It has been observed that *Vitis riparia* contains more malic acid than tartaric acid [7], and that other Vitis species tend to have more tartaric acid than malic acid. In grape must, depending on the content and form of organic acids, the pH varies from 2.9 to 3.8. These variations of pH induce a change of color in red musts, because red pigments from grape skins, the anthocyanins, have pH-dependent hues. At a grape must and wine pH, anthocyanins are predominantly present in the flavylium cation form, but the chalcone form is also present [20]. The structure of anthocyanins is highly different between Vitis species, as *V. vinifera* grape skins are rich in anthocyanin mono-glucoside, especially malvidin-3-*O*-glucoside, while interspecific hybrid grape skins are composed of anthocyanin mono- and di-glucosides [21]. Anthocyanins are found in the vacuoles of the cells in the same location as condensed tannins and monomeric flavanols. Those compounds can interact through a co-pigmentation phenomenon and form pigmented compounds, such as pigmented tannins or polymeric pigments [21,22,23]. The formation of anthocyanin-tannin compounds is responsible for the color stabilization in wine made from *V. vinifera* grapes, but little is known about the reactivity of tannins with anthocyanins from hybrid grapes. Burtch and Mansfield [21] suggested that wine from hybrid grapes is less susceptible to form polymeric pigments, even though the anthocyanin content is much higher than in wines from *V. vinifera* grapes.

## 3. Chemistry of Tannins

### 3.1. Hydrolyzable Tannins

Hydrolyzable tannins are readily hydrolyzed by acids, bases, hot water and some enzymes. Two groups of hydrolyzable tannins exist: gallotannins and ellagitannins. Their name comes from either the gallic acid or ellagic acid unit obtained after hydrolysis.

Gallotannins have a core structure of either glucose or less commonly shikimic acid or quinic acid, which is esterified by up to five gallic acids (Figure 1A). 1-galloyl-β-d-glucose (syn. Glucogallin) is formed in plant by a glucosyltransferase that catalyzes the esterification of UDP-glucose and gallic acid. Glucogallin then acts as an acyl donor and acceptor in the biosynthesis of β-1,6-digalloyl-glucose, β-1,2,6-trigalloyl-glucose, β-1,2,3,6-tetragalloyl-glucose, and β-1,2,3,4,6-pentagalloyl-glucose. Galloyl groups may be further esterified by gallic acid through depsidic bonds and up to thirteen groups, i.e., trideca-galloylglucose, which has been found in Chinese sumach tannin extracts [24,25]. In Tara extracts, 5-mono-galloyl-quinic, 4,5-di-galloyl-quinic, 3,4,5-tri-galloyl-quinic and 1,3,4,5-tetra-galloylquinic have been found as the core structure that can be esterified by more gallic acids via depsidic bonds [26,27]. Galloylshikimic is much less common, but the 3-*O*-galloylshikimic has been identified in *Erodium cicutarium* herb [28].

Ellagitannins are widely found in fruits, seeds, wood, etc., and are the product of oxidation of penta-galloylglucose (Figure 1B). The hexahydroxydiphenoyl (HHDP) and nonahydroxydiphenoyl (NHDP) moieties formed by oxidative biaryl-coupled (C-C coupling) between galloyl residues, are esterified to an open-chain glucose in position 4 and 6 and in position 2, 3 and 5 respectively (Figure 1B) [29,30,31]. The most common ellagitannins are castalagin and vescalagin found in oak barrel [32]. Ellagitannins tend to form dimers in solution called roburins found in wine aged in oak barrel. As these compounds are readily hydrolyzable, castalin, and vescalin, which are the NHDP moiety esterified to an open-chain glucose, as well as ellagic acid, can be released after the hydrolysis of castalagin and vescalagin respectively.

### 3.2. Condensed Tannins

Condensed tannins or proanthocyanidins are oligomers and polymers of flavan-3-ol. Flavanols are composed of a carbon backbone of C6-C3-C6, and are comprised of two aromatic rings A and B and a pyran ring, the heterocycle C [33]. Monomers of flavan-3-ols are commonly called catechins, and can be distinguished by the stereochemistry of the asymmetric carbons C2 and C3, the presence of galloyl groups, and the level of hydroxylation on the B-ring. Di-hydroxylation at C3′ and C4′ of the B-ring is a catechol type which corresponds to (+)-catechin and (−)-epicatechin, and the tri-hydoxylation on the B-ring corresponds to (+)-gallocatechin and (−)-epigallocatechin. The 2R configuration at C3 is more common than the 2S [3]. The condensation reaction occurring between the carboxy group of a gallic acid and the hydroxyl group on C3 of catechins generates catechin gallates, such as (−)-epicatechin-3-*O*-gallate and (−)-epigallocatechin-3-*O*-gallate found in tea and grapes. The so-called proanthocyanidins have the ability to release anthocyanidins after cleavage of interflavan bonds under acidic and oxidative conditions [33,34,35]. Thirteen categories of proanthocyanidins are known, based upon their level of hydroxylation on the A- and B-ring, as well as at C3 [33]. Proanthocyanidins or condensed tannins are characterized by the nature of the constitutive units, i.e., the nature of the monomer of flavanols, by the type of interflavan bonds binding the monomers, and by the average number of the constitutive units, i.e., the mean degree of polymerization (mDP) (Figure 1C–E).

The nature of the constitutive units varies according to the source, such as (+)-catechin and (−)-epicatechin that are commonly found in fruits, while (+)-gallocatechin and (−)-epigallocatechin are more common in leaves [36,37]. Constitutive units include extension units that are monomers connected to two other units, and terminal units, which are connected to only one unit in the molecule. In some fruits, such as apple, the (+)-catechin is only found as a terminal unit while (−)-epicatechin can be found as extension and/or terminal units. (−)-epicatechin is the most abundant constitutive unit found in fruits [38,39]. Flavanol monomers are linked through carbon–carbon linkages between the pyran C-ring of one monomer and the aromatic A-ring of another monomer, either in C4-C8 or C4-C6 fashion called B-type (Figure 1C–E), or in C4-C8 or C4-C6 B-type linkage with an additional ether linkage at C2-O-C7 or C2-O-C5, called A-type [37]. The mDP corresponding to the average number of constitutive units can provide an estimation of the molecular mass of proanthocyanidins after an acid–catalysis reaction in presence of a nucleophile. By heating the acidic medium, the linkages are cleaved, and a carbocation is formed at the C4 of the extension unit, while the terminal unit is released. The carbocation then reacts with the nucleophile, either a benzylthioether or phloroglucinol or thioglycolic acid, to form an extension unit linked to a nucleophile. After reaction, the extension units and terminal units are identified and quantified to calculate the mDP, and then used to estimate the molecular mass of proanthocyanidin, based on the molecular mass of the constitutive units. So far, this method is a commonly used method for the characterization of constitutive units, concentration, and mDP of condensed tannins, but it only provides an estimation, as the reaction is never complete depending on the conformation of the tannin, as well as the type of linkages, such as A-type linkages that are more resistant to the acid-catalyzed reaction. It has also been pointed out that the nucleophile might react in other positions, rather than C4 and that oxidized proanthocyanidins and pigmented tannins, which are tannins bound to anthocyanins, do not react the same way with the acid-catalysis and the nucleophile. Recently, Zeng et al., [40] discovered a new cyclic procyanidin tetramer, which did not release a flavan-3-ol terminal unit after acid-catalysis in presence of phloroglucinol. This “crown procyanidin tetramer” is composed of four epicatechin units linked in C4-C8 and C4-C6 [40] (Figure 1E).

## 4. Tannins in Grape and Wine

### 4.1. Biosynthesis of Tannins in Grape

Condensed tannins are biosynthesized in plants through the phenylpropanoid pathway that also produces hydroxycinnamic acids, flavones, flavonols, and anthocyanins. Flavan-3-ol monomers and anthocyanidins, the anthocyanin aglycone, are synthesized from flavan-3,4-diols (leucoanthocyanidins). The leucoanthocyanidin reductase converts the leucocyanidin into catechin and an epimerase converts it to epicatechin. The epicatechin monomer is also produced from the conversion of cyanidin through an anthocyanidin reductase. The polymerization of flavan-3-ol is still not identified and two hypotheses are suggested: a non-enzymatic mechanism by which flavan-3-ol monomer is added to another unit by nucleophilic displacement [41], and an enzymatic mechanism involving a polyphenol oxidase catalyzing the condensation of monomers into oligomers and polymers [42,43]. The concentration of condensed tannins in grape berries is positively correlated to the level of leucoanthocyanidin and anthocyanidin reductases. In general, skin and seed tannins accumulate from flowering to véraison, and then their concentration decreases during berry maturation, due to either a reduction of their extractability resulting from the reaction of tannins with proteins, polyphenols, and polysaccharides from the cell walls and/or oxidation reactions [44,45,46].

### 4.2. Tannins in Grape

The concentration of condensed tannins in grapes varies according to the variety and the ripening conditions.

#### 4.2.1. *Vitis vinifera*

In *Vitis vinifera* grapes, condensed tannins are found in stems, skins, flesh and seeds but their chemical structure, concentration, and mDP varies depending on the berry parts and the variety. Only some precursors of hydrolyzable tannins such as gallic acid, ellagic acid, galloyl-glucose, and di-galloyl-glucose have been found in *Vitis vinifera* grapes [4].

The bunch stems of Castelao Frances and Touriga Francesa contain at harvest between 28 and 35.8 mg of proanthocyanidin/g of stem fresh weight [47]. In Merlot stems, tannins represents about 5 g/kg of stem fresh weight [5]. Stem tannins are composed of (−)-epicatechin as the main extension unit and (+)-catechin as the main terminal unit. A very low proportion of (epi)gallocatechin was found in the Merlot grape stem tannin only as an extension unit, and the proportion of gallates was close to 15% in these stems [5]. The mDP of tannin in stems varies from 5 to 9 [5,47].

Condensed tannins from *V. vinifera* grape skins accounts for about 3 to 110 mg/g berry dry weight and are composed of (-)-epicatechin as the main extension unit, followed by (−)-epigallocatechin, (−)-epicatechin-3-*O*-gallate and (+)-catechin as extension units and as terminal units (Table 1). The (−)-epicatechin and (−)-epicatechin-3-*O*-gallate were not found as terminal units in Syrah and Merlot skins [22,48]. During berry ripening, the proportion of (−)-epigallocatechin as an extension unit increased and the (+)-catechin as the terminal unit decreased. This is associated with an increase in the mDP from 4.5 in green berries to 27 in red berries. The mDP of skins has been calculated to be about 33 in Merlot and 31 in red Shiraz berries [48,49]. Recently, a crown procyanidin tetramer composed of (−)-epicatechin has been identified in wine, and only in skins of Cabernet sauvignon, and was absent from seeds and the bunch stem [40]. 

In *V. vinifera* grape seeds, the condensed tannin concentration is much higher than in skins varying between 11 to 140 mg/g berry dry weight in Cabernet sauvignon and Merlot, and tend to be stable during berry ripening from véraison to harvest maturity. In contrast with the constitutive units in grape skins, epigallocatechin as an extension unit was not found in any *V. vinifera* grape seeds. Epicatechin was also the main extension and terminal unit found in seeds, and the proportion of these subunits varied during berry ripening. The proportion of (−)-epicatechin as a terminal unit tend to decrease during ripening from veraison to harvest maturity, while the (+)-catechin as a terminal unit increases in Syrah grapes but the opposite has been observed in Cabernet sauvignon grapes [50,51]. The mDP in *V. vinifera* grape seeds varies between 3.8 and 11 in Cabernet sauvignon at harvest maturity and in Syrah after véraison [50,51]. During grape maturation, the mDP of tannins in grape seeds decreases starting with a maximum of 5.9 at fruit-set in Cabernet sauvignon to a minimum of 3.8 at harvest maturity [51]. 

These variations of tannin composition and concentration in *V. vinifera* grapes have been attributed to variety, climatic conditions, and viticultural practices such as leaf removal post fruit-set, which increases the temperature of the berry and leads to a higher production of condensed tannins in Merlot grapes [52].

#### 4.2.2. Hybrids

The concentration of tannins in V. spp. grape seeds and skins has been measured by the protein precipitation method using bovine serum albumin and compared to *V. vinifera* grapes (Table 2) [55,56]. 

The total tannin concentration (skin + seed) in grapes from V. spp. was much lower than in *V. vinifera* grapes (0.71 mg/g berry versus 1.27 mg/g berry as (+)-catechin equivalent in French–American hybrid grapes, such as Baco noir, Maréchal Foch, Leon Millot, and De Chaunac and in *V.vinifera* grapes such as Pinot noir, Cabernet sauvignon, Merlot, and Cabernet franc, respectively). Similar concentrations were observed in Frontenac, Marquette, and St Croix grapes (0.49 mg/berry as (+)-catechin equivalent) [56]. The concentration of tannins in seeds was much lower at harvest than at véraison, but was higher than in skins from French–American hybrid grapes (about 0.20 mg/g berry of skin tannins and 0.60 mg/g berry of seed tannins). This is in comparison to about 0.50 mg/g berry of skin tannins and 1.40 mg/g berry of seed tannins in *V. vinifera* grapes. These concentrations varied significantly depending on the grape variety, e.g., Maréchal Foch contains a higher grape skin and seed tannin content than Frontenac.

Even though some studies have focused on tannin content in Vitis spp., little is known about the chemical structure of those tannins. Curko et al., 2012 [58] compared the composition of proanthocyanidins in seeds from *V. vinifera* grapes and American Vitis spp., and observed that the percentage of galloylation of seed tannins in *V. doaniana* and *V. champinii* was significantly high. The flavanol monomers in American Vitis spp. grape skins were identified as catechin gallate and epicatechin gallate and the tannins had a higher percentage of galloylated forms (~3%) than in *Vitis vinifera* grape skins [4]. The mDP of tannins in American spp. grape skins has been reported to vary between 4 and 17. In grape seeds, the differences in mDP varied between 5 and 12 in American and hybrids but were more comparable to the mDP in *Vitis vinifera* grape seeds [4]. In grape seeds from some hybrid grapes such as Seyval, only traces of dimer-gallates and trimers have been observed compared to other hybrids [59]. More recently, dimeric and oligomeric procyanidins were found in lower quantities in *Vitis cinerea* compared to in *Vitis vinifera* grape seeds, but not as low as in American Vitis grape seeds. 

In comparison to *Vitis vinifera* grapes, wild American grapes contains 1000 times higher ellagic acid, a precursor of hydrolyzable tannins in their skins. Narduzzi and co-workers [4] identified for the first time the presence of gallic acid, galloyl-glucose as well as oligomeric hydrolyzable tannins including HHDP-galloyl-glucose in wild American grape skins.

The composition and concentration of tannins in *Vitis vinifera* and other Vitis species are highly different in skins compared to seeds, and tend to be lower in Vitis interspecific hybrid varieties. Therefore, the concentration and composition of tannins found in wines is usually lower in Vitis spp. than in *Vitis vinifera* wines.

### 4.3. Tannins in Wine

Condensed tannins are found mostly in red wines because of the maceration time and skin contact that is typically employed in red wine making over white wine making, providing a better extraction of tannins from skins and seeds to the wine. The tannin concentration in red wines varies from 50 mg/L in wine from *V. riparia* and *V. labrusca* grapes to 4 g/L in wine from *V. vinifera* [60] (Table 3). In some dry white wines, a concentration of tannins from 100 to 300 mg/L can be found, which can be related to the quality of juice settling.

As previously shown by Aron and Kennedy [64], the extraction of proanthocyanidins increased significantly during alcoholic fermentation, e.g., the tannin content in Pinot noir must and wine increased from 200 to 1000 mg/L after 6 and 21 days of fermentation, respectively. Similarly, the mDP of tannins extracted from must after 6 days of fermentation was 3.89 and increased to 5.89 after 21 days of fermentation. In Table 4, a summary of the effect of the winemaking process on the content and size of tannins from *V. vinifera* and interspecific hybrid wines is shown. Skin tannins are readily extracted due to skin breaking, and the presence of alcohol during fermentation and maceration. It leads to a diffusion and an extraction of tannins and anthocyanins from grape skins into the alcoholic medium, wine [39,65]. Ethanol produced by yeasts during alcoholic fermentation from grape sugars lead to a reorganization and solubilization of the grape seed lipids, which favor the extraction of tannins from seeds. This latter extraction takes more time and is slower than the extraction of tannins from skins due to the physical structure of seeds [65]. It has been previously observed that in wine the chemical structure of tannins are epigallocatechin-rich tannins extracted from grape skins, rather than epicatechin-3-*O*-gallate-rich tannins from seeds [66]. The extraction of tannins into wines is also highly dependent on their molecular weight, related to their solubility, as well as the cell wall material including polysaccharides and proteins surrounding them in grape cells. 

#### 4.3.1. *Vitis vinifera*

In *V. vinifera*, the tannin content in wines increase with the increase in alcohol level, temperature, the addition of enzymes and the combination of some of those parameters. The level of tannins in red wine made from 50% destemmed clusters was higher than wine made from 100% destemmed clusters, showing that tannins from stems can be extracted during maceration [69].

Procyanidin dimers tend to be extracted from grape skins and seeds during maceration and alcoholic fermentation after 3 to 4 days, as observed in Tempranillo musts and wines by Berrueta et al., 2020 [70]. The extraction of tannins increases with an increase in the maceration period, due to a longer skin contact time. In Cabernet sauvignon, Pinot noir, Merlot, Syrah, and Tempranillo, an increase in skin contact time from 4 days to 10 days or 36 days increased the levels of total polymeric phenols and tannins, as well as the extraction of larger molecules of tannins [70,71,72,73] (Table 4). As previously explained, tannins extracted from grape skins have a larger proportion of epigallocatechin compared to epicatechin-3-*O*-gallate, which is mostly extracted from grape seeds, and the skin tannins are commonly extracted first followed by the seed tannins. 

Thermovinification [74,75] and flash release [68,76,77] techniques have also been examined in terms of their enhancement of tannin extraction. Thermovinification, along with pulsed electrical-field and enzyme treatment, were all shown to have higher rates of phenolic extraction over a control fermentation [74]. Overall, the thermovinification treatment had a higher concentration of phenolics and specifically flavonols in must, but did not show significant increase in tannin content in wines. For these experiments, thermovinification treatment was performed on crushed grapes at 70 °C for 30 min. Flash release treatment, which combines thermovinification treatment, followed by an ultra-low-pressure environment, has also been shown to have a major impact on tannin extraction. A report from Morel-Salmi et al. demonstrated that this treatment caused an increase in proanthocyanidin concentration for several cultivars (Grenache, Mourvèdre and Carignan), with the mDP reported to be very similar between control and treated wines [68] (Table 4). Further experiments into the effect of the thermovinification portion of the treatment revealed that a longer heat treatment (9 min at 95 °C vs. 6 min at 95 °C) increased the extraction of proanthocyanidins into the must, however, this result was short-lived, and after 1 day of fermentation, the proanthocyanidin concentration was the same for both flash release treatments. Pinot Noir and flash release treatments have also been investigated [76]. Overall, there was an increase in flavan-3-ol concentration in must and directly after fermentation for treated samples. However, after a maturation in bottle at 4 °C for 6 months, the flash release treated wines had less flavan-3-ol content than the control. Despite this result, the authors claim that the amount of flavan-3-ols extracted (including monomers and proanthocyanidins) is greater than previously reported to be extracted from Pinot Noir wines.

Non-thermal methods have been also used on *Vitis vinifera* musts to improve tannins extraction. Microwave-assisted extraction is a method commonly used in the food industry that showed an improvement of tannin extraction in Pinot noir wines, but not in Cabernet sauvignon or Merlot wines [78,79]. In a recent study carried out by Cassassa et al., 2019 [63] on the use of microwaves for the extraction of tannins in unripe, ripe, and fully ripe Merlot grapes, this technique showed a higher tannin content in wines after microwave-assisted extraction compared to the control, but the phenolic extraction was higher in unripe grapes. These results were similar to some thermovinification treatments, suggesting that cell wall materials are affected by those treatments, which likely reduces the extraction of tannins.

#### 4.3.2. Hybrids

In Brazilian red wine made from Maximo hybrid grapes (a cross between *V. vinifera* cv. Syrah and Seibel 113432), the total level of phenolic compounds was higher than in the wines made from *V. labrusca* grapes, such as Isabella or Ives [61]. Similar results for the concentration of tannins in French–American hybrid wines were observed by several researchers [23,57,62]. The concentration of tannin in Baco noir, De chaunac and Maréchal Foch, as well as in Marquette and Frontenac, was much lower than in *V. vinifera* wines (Table 3). Nicolle et al., 2019 [80] observed that in red wine made from Frontenac grapes, the tannin concentration varied from 149.2 mg/L (+)-catechin equivalent after 4 days of maceration to 217.7 mg/L (+)-catechin equivalent after 15 days of maceration when measured by the protein precipitation method. 

The lower concentration of tannins in wine has been attributed to the ability of tannins from grape skins and seeds to bind with proteins and cell wall material. The reactivity of tannins will be presented in the next section. Some studies focused on the application of winemaking treatments already used on *V. vinifera* musts, in order to improve the tannin content in hybrid wines. However, as a result of a lack of knowledge on the tannin chemistry from those V. spp. grapes, the advantages of the different winemaking techniques have not been optimized.

Manns and co-workers [23] processed hybrid grape juice with different traditional treatments, such as the addition of bentonite, heating, and flash freezing, and compared these with *V. vinifera* (Table 4). A decrease in the protein content was observed for all those treatments, but it was not related to any increase in the tannin concentration in Maréchal Foch wines in contrast to Cabernet franc. The use of macerating enzymes (pectinase and protease) after crushing on Maréchal Foch musts did not show any increase in its tannin content and mDP (between 2.8 and 3.0). The same authors studied the effect of the addition of exogenous grape tannin extract on the tannin content and size in hybrid red wines and observed a low increase in tannins in Maréchal Foch wines, with no change of the mDP after the addition of a high concentration of exogenous tannins. This could be explained by the very low percentage of tannins contained in commercial grape tannin extracts (maximum up to 38% by weight) [81].

Norton and co-workers [62] hypothesized that the concentration of tannins in hybrid red wines might be increased by blending—before fermentation—a low-tannin hybrid cultivar-Marquette, with a high-tannin *V. vinifera* cultivar-Cabernet sauvignon. However, the final tannin content was lower than predicted, potentially due to the high level of proteins in those wines. Due to a lack of information about the composition of tannins in grapes and wines from V. spp., it is still difficult to understand the reason of a decrease in tannin content in grape skins during berry ripening, and of an increase in the concentration during winemaking. Tannins can be highly reactive with cell wall material, including polysaccharides and proteins from grapes, as will be discussed in the section below. Hypotheses on this lower tannin content in wines from hybrids include:(1)A lower solubility of tannins due to their chemical structure, as it has been previously shown that oligomeric tannins tend to form aggregates and be less soluble than polymeric tannins [82].(2)A lower tannin extraction and higher retention in grape skins either due to:
-the reactivity of tannins with cell wall material in grapes reducing their extractability for quantitative and qualitative analysis [83], or-the reactivity with plant enzymes or other proteins [84], or other phenolic compounds, such as anthocyanins, commonly mono- and di-glucosides in hybrid grapes [21].


## 5. Reactivity of Tannins

### 5.1. Interactions between Macromolecules and Tannins in Vitis vinifera and Interspecific Hybrids

During the growing season and fruit development, polyphenols in the cell vacuoles are separated from other macromolecules, such as polysaccharides and proteins in the grape cell walls. This implies that any attractive interaction occurs during or after extraction of the tannins into the must/wine matrix with polysaccharides, proteins that are soluble within that matrix, or complexed to remaining cellular structures. The reactivity of tannins from *V. vinifera* wines with salivary proteins, such as proline-rich proteins, has been extensively studied, as it is the first step in the mechanism of astringency that commonly leads to the mouth drying sensation of red wines [85,86,87,88,89,90,91]. Recent research has also focused on the interactions between cell wall material (CWM) and tannins in *V. vinifera*, to improve the understanding on the extraction or retention of tannins during winemaking. CWM is comprised of both polysaccharides and proteins, is cultivar dependent, and is challenging to fully characterize [92,93]. For the interspecific hybrids, most work has focused specifically on protein interaction with tannin, since these cultivars have been shown to have a large concentration of soluble protein compared to *V. vinifera* (hybrid juice-175.75 mg/L, *V. vinifera* juice-146.2 mg/L; hybrid wine-93.96 mg/L, *V. vinifera* wine-15.95 mg/L) [57]. 

Non-covalent interactions between tannins and other macromolecules commonly involve hydrophobic interactions and hydrogen bonds (Figure 2), as well as electrostatic interactions, depending on the structure of the macromolecules and the environmental matrix. π-π stacking also exists between the aromatic rings of polyphenol and protein, but has a much lower binding energy and is therefore not as common as hydrogen bonds or hydrophobic interactions (Figure 2). Hydrophobic interactions are described as the strong attraction of hydrophobic surfaces and groups in water [94]. Hydrogen bonding occurs when a hydrogen atom, covalently bonded to an electronegative atom, interacts non-covalently, but quite strongly with a separate electronegative atom [95]. Most proteins and tannins have multiple binding sites where these types of interactions are possible, and, similarly to hydrophobic interactions, their respective structures will influence the strength and number of these types of bonds.

### 5.2. Effect of the Structure of Macromolecules

Plant cell walls are characterized as a network of polysaccharides with proteins, and some minor compounds such as phenolic acids and minerals. This network provides the rigidity and elasticity to plant cells during their development and includes cellulose microfibrils, hemicelluloses, and pectins as the polymers of saccharides and some extensins: glycoproteins associated with pectins. In order to improve the understanding of the interactions between tannins and cell wall material in grapes and wine, model studies from Le Bourvellec and Bindon groups with extracted CWM and purified tannin from apples and grapes have been used [46,96,97,98,99]. The apparent affinities of specific polysaccharides from CWM, as well as starch, have also been investigated, with procyanidins extracted from apple, pear, and grapes with different structure and size. Starch and pectins showed the highest affinities with highly polymerized and galloylated procyanidins. The apparent affinities between those polysaccharides and procyanidins were ranked as follows: pectin >> xyloglucan > starch > cellulose [97,99,100].

Pectins tend to form a gel type network, and it was hypothesized to have strong interactions with the procyanidins, due to a large number of hydrophobic regions and the “hydrophobic cavities” formed. The overall structure and conformational organization of the polysaccharides was suggested to play a large role in the apparent affinities (pectin-gel with hydrophobic cavities, cellulose-microfibrils, xyloglucans-globular, starch-porous granules). A more in-depth look into the effect of the pectin structure and procyanidins from apples has also been reported [99]. Pectins are very complex polysaccharides from plant cell walls that are composed of two main regions: the smooth, also called homogalacturonans and the hairy, including rhamnogalacturonans type I and type II. Homogalacturonans are characterized by a backbone of galacturonic acids more or less acetylated and/or methylated. Watrelot et al., 2013 [99] have shown that highly methylated homogalacturonans (degree of methylation of 70%) had the highest affinity for larger procyanidins, due to hydrophobic interactions and hydrogen bonds. In contrast to smooth regions, hairy regions, specifically rhamnogalacturonans II, had much lower affinities for procyanidins, due to the lateral side sugar chains limiting the accessibility of procyanidins into the “hydrophobic cavities” of the pectins [100].

During fruit development and ripening, the network, the physical state, and composition of polysaccharides from CWM change, as well as their ability to interact with tannins. In over-ripened pears, the pectic hairy regions lose their lateral side chains, which increases the porosity of flesh cell walls, and leads to a higher affinity with procyanidins [101]. These authors observed that the strength of associations of CWM from different fruit parts with tannins decreases as follows: Parenchyma cells > Flesh > Stone cells > Skins. Bindon et al., 2012 [102] also observed that the affinity for proanthocyanidin to skin CWM increases during grape ripening, as the skin CWM may change configuration during ripening, with an increase in porosity and other molecular restructuring. The same authors [103] examined the extraction of proanthocyanidins from different grape sources (skins vs. seeds) when mesocarp was included or not during the extraction. They concluded that mesocarp preferentially bound seed tannin over skin tannin, and that anthocyanins influenced the extraction of tannins. The authors suggested that a higher juice: solids ratio for red wine fermentations may be beneficial to increase the overall extraction of tannins by minimizing effects of a concentration gradient of tannin. 

As mentioned above, the plant cell wall materials are also composed of proteins that can interact with tannins during processing, but extensive research on the affinities between proteins and tannins has been carried out on proline-rich proteins (PRPs) commonly found in saliva, as they are highly related to the red wine astringency perception. It has been shown that proline residues provide interaction points with polyphenols through hydrophobic interactions and hydrogen bonds, similar to CWM-tannin interactions (Figure 2). Aromatic rings of tannins can interact through hydrophobic stacking with the pyrrolidine ring of proline and form hydrogen bonds with the preceding amino acid amide bond [2,104]. The proline residue also allows the protein to retain an open conformation by preventing hydrogen bonding between amino acid residues of the protein [105,106]. 

Basic amino acids carry positive charges at the pH of grape juice and wine. These positively charged residues, lysine, histidine, or arginine, can interact with tannins through electrostatic interactions. While it has been suggested that hydrophobic interactions play a large role in the association of tannins and proteins, histidine-rich proteins from saliva have been shown to more strongly interact with tannins than PRPs [107,108]. Protein conformation also plays a role in tannin-protein interactions. The affinity between a helicoidal protein (poly-l-proline) and a procyanidin of an mDP of 8 was higher than with a globular protein, bovine serum albumin [109]. However, under high tannin concentration or with large tannins, the interaction with globular proteins can be significant [105]. 

Proteins in grapes are present in the cell cytoplasm, as well as part of the cell wall structure. The main proteins present in several hybrid cultivars have been identified as chitanase, thaumatin-like protein, and β-endoglucanase, which are classified as functional and necessary for metabolism and energy production [84]. These proteins are classified as pathogenesis related proteins, and are also the proteins implicated in white wine haze of *Vitis vinifera* [55,110,111]. A recent report identified over 100 proteins in Sauvignon Blanc grapes following extraction, trypsin digestion and LC-MS/MS experiments [112]. While there was some overlap with the location of certain proteins in respect to the skin, pulp, and seed tissues, the majority of identified proteins were found in both skins and pulp (38), in pulp and seeds (15), and in skins and seeds (11).

### 5.3. Effect of the Structure of Tannins

Several variables of tannin structure are important in understanding interactions between tannins and macromolecules. These include types of constitutive units, the types of linkages connecting these units, the mean degree of polymerization (mDP) or size of the tannins, and conformation of longer chain tannins. In grapes, these variables depend on grape species and cultivar, and can be affected by the conditions of the growing season as previously explained.

The stereochemistry of the pyran ring of the flavan-3-ol units has an effect on the binding potential and aggregation of tannins to proteins [113,114]. A study showed that monomeric (+)-catechin, epicatechin-3-*O*-gallate, and epigallocatechin-3-*O*-gallate produced an aggregate haze when exposed to poly-l-proline, however, monomeric (−)-epicatechin and (−)epigallocatechin did not [87]. In this experiment, the (+)-catechin concentration needed to be significant, in order to allow for soluble complexes to aggregate through interaction of the exposed flavan-3-ol unit and become insoluble. The galloylated monomers are suggested to have the ability to interact with two protein units each in a bilateral fashion, and subsequently cause aggregation, similarly to (+)-catechin. The reason that the non-galloylated monomers did not cause an insoluble aggregate to form, was hypothesized to be: (1) the interaction did not cause insolubilization of the complexes; or (2) the non-galloylated monomers were more soluble than the galloylated monomers, and therefore interacted less with the proline-rich proteins. Isothermal titration calorimetry was also used to evaluate these monomers, along with oligomers with a mDP of 3.85 with poly(l-proline) [115]. The larger oligomers were observed to have a higher association constant with the salivary protein, which was explained by a larger entropic contribution. This change in entropy comes from the increase in hydrophobic interactions and the loss of water molecules on the protein surface. This is in agreement with previous studies that showed that larger tannins will have a greater affinity to fining proteins [116]. 

Also important is the type of interflavan linkage: C4-C8 vs. C4-C6. Dimers with C4-C8 linkages were shown to have higher activity towards protein than dimers with C4-C6 [113]. The greater affinity for C4-C8 linked oligomers is postulated to be due to improved conformation for favorable interaction over the C4-C6 linked oligomers.

In general, there have been several reports that confirm that CWM adsorbs proanthocyanidins with a higher molecular weight [46,117]. One report, however, stated that, as the length of the proanthocyandin chain increased the affinity for skin derived CWM decreased [96]. When examining CWM and the adsorption of proanthocyanidin, a large body of work has been published from Bindon and co-workers at the AWRI [46,83,96,102,103,117,118,119,120,121]. While many of these experiments concluded that CWM derived from the flesh component of grapes had a higher affinity for proanthocyanidin adsorption, it was also observed that the larger the proanthocyanidin, the higher the affinity for the CWM. Even though skin proanthocyanidins typically have a larger mDP, contrary to the previous statement, the Bindon team also observed a higher affinity for seed proanthocyanidin for CWM (of flesh or skin) [117]. The seed proanthocyanidins that were adsorbed were of a higher molecular weight, supporting the first study.

The size of tannins or mDP is positively correlated with protein precipitation, with molecules of higher mDP having an increased ability to precipitate proteins [105]. The reason behind this is due to the increase in the number of functional groups available on the tannin to interact with a protein. For example, the tannin-protein reactivity of procyanidin dimer B3 with commercial BSA is double that of the monomer (+)-catechin [113]. In the same line of reasoning, increasing the degree of galloylation of tannins has been shown to have an increase in protein precipitation. By enhancing the hydrophobic interactions with increased π-π interactions, the complexes can interact more strongly (Figure 2). Examples of this include experiments where (−)-epicatechin or (−)-epigallocatechin do not precipitate with salivary proteins or gelatin, however, (−)-epicatechin-3-*O*-gallate or (−)-epigallocatechin-3-*O*-gallate will cause precipitation with the aforementioned proteins [122]. It has also been shown that increasing the number of hydroxyl groups on the flavan-3-ol backbone (i.e., from (−)-epicatechin to (−)-epigallocatechin) will increase the interactions and overall binding to proteins [123].

### 5.4. Effect of the Grape and Wine Matrix

The environment of a wine fermentation has a great impact on the extraction and retention of tannins from red grape skins and seeds. Some chemical and physical factors contribute to the extraction of tannins, including pH, ethanol concentration, and temperature. Researchers have also examined the physical or chemical removal of CWM and proteins, in order to enhance tannin extraction and retention in both *V. vinifera* and hybrid cultivars. Since different grape cultivars can have very different concentrations of matrix components, it is difficult to develop a defined protocol for all fermentations, in order to control or maximize tannin extraction.

It has been established that proteins have a stronger affinity for tannins when the pH of a solution is near the pI of the protein [124]. A recent example showed that pH had a significant effect on the ability of protein and tannin to interact and precipitate [125]. Protein-tannin complexes precipitated more readily at a pH that corresponded to the pI of the protein (the globular protein-Bovine Serum Albumin was under investigation). This result is consistent with the understanding that the solubility of proteins is lowest at the pI, and that the 3-dimensional structure is weakened, leading to a more open structure that is available for increased interactions.

Solubility of tannins is important in a grape or wine matrix, due to changing concentrations of ethanol. The effect of ethanol on the extraction of tannins from grapes to wine has been explained above, but its effect on protein-tannin interactions is not fully understood. It has been shown that the higher the solubility of tannins in water, as compared to n-octanol, the weaker the interaction with proteins, due to the lack of hydrophobic interactions [82,126].

A series of reports have been published in support of a model developed for the extraction of phenolics from red grapes during winemaking, based on experimental data [127,128,129,130,131]. The model was developed as a function of temperature and ethanol concentration, and takes into consideration the release of phenolics, the adsorption of phenolics onto grape material, and the decrease in anthocyanin over time. In a study supporting this model, it was found that increasing both temperature and ethanol concentrations increased the equilibration rate for the adsorption of proanthocyanidin to CWM, as well as a preference for large molecular weight proanthocyanidins across all conditions [131]. A subsequent report suggested that proanthocyanidin-CWM interaction may be irreversible at lower temperatures, and reversibly adsorbed/desorbed at higher temperatures [131]. The researchers noted that the larger molecular weight proanthocyanidins were more readily desorbed. An early report in this collection of work investigated the temperature influence of the cap and must during a Cabernet sauvignon fermentation on tannin extraction from skins or seeds [127]. Skin and seed phenolic compounds were extracted differently under increased temperature conditions. The skin tannins were extracted at a higher rate, but overall, the concentration did not increase, whereas the seed tannins were extracted at a higher rate and did see increases to their overall extraction. Interestingly the increase in must temperature was found to have a greater effect than the increase in cap temperature.

## 6. Conclusions

Based upon previous research of *V. vinifera* varieties, winemakers are applying different methods to extract more tannins during red winemaking, especially in hybrid wines, either by removing CWM, or proteins.

A recent report using Monastrell grapes saw the grapes crushed, pomace separated, juice settled (similar to white or rose pre-fermentation), pomace re-introduced, and maceration/fermentation carried out under standard conditions [132]. This study aimed to remove excess CWM and polysaccharides from the fermentation. The study reported a 43% increase in tannin, as compared to a control fermentation with no settling of juice. The types and sizes of tannins were concluded to be similar in both wines, although there was a greater quantity of them in the wine produced from settled juice.

Another set of experiments performed on Frontenac grapes evaluated several winemaking techniques to promote tannin extraction by reducing interactions with CWM and proteins [80]. Must was heated or bentonite treated to remove protein, as well as pomace inclusion or exclusion during the fermentation. Wines were also macerated for an extended period of time, in order to allow more tannin to be extracted. The authors concluded that cold-maceration prior to fermentation, followed by pomace removal during fermentation, is beneficial for tannin retention for this hybrid, however they also suggested that a 3 g/L addition of enological tannins to heat treated must was also necessary, to improve on the final wine’s astringency.

In this review, we have highlighted the differences between tannins in grapes and wine made from *V. vinifera* or interspecific hybrids. The variability of tannins due to species, variety, climate, and viticultural practices makes understanding tannin behavior in grapes and wine difficult. Further work into the composition of the CWM of interspecific hybrids is necessary to understand if and how these components are involved in the reduced extraction and poor retention of tannins in the subsequent wines. Future studies should focus on both *V. vinifera* and interspecific hybrid red cultivars, identifying and enumerating the polysaccharides and proteins present in grapes and wines, and determining what impact they are specifically having on tannin extraction during fermentation and retention in finished wines. In the case of interspecific hybrids where there is good correlation between low wine tannin and high wine protein concentrations, mitigation techniques need to be investigated further, to allow winemakers to maximize tannin extraction without sacrificing other factors of wine quality.

## Figures and Tables

**Figure 1 molecules-25-02110-f001:**
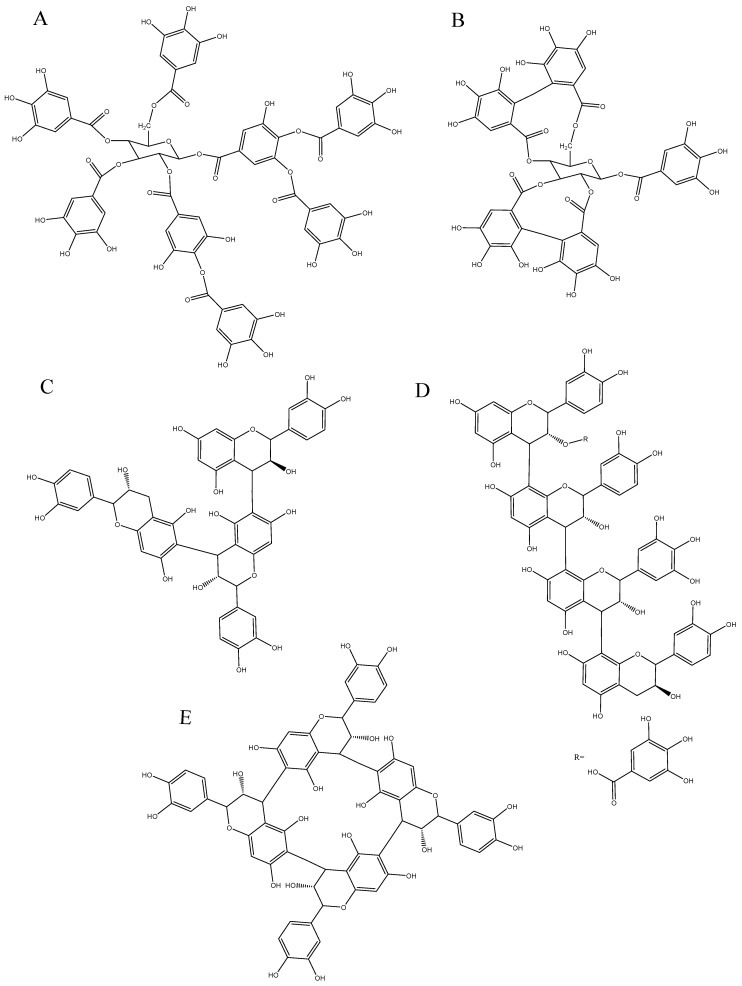
Chemical structure of hydrolyzable tannins, gallotannins (**A**); and ellagitannins (**B**); and condensed tannins including trimer of catechin and epicatechin linked in C4-C6 (**C**); tetramer composed of epicatechin gallate, epicatechin, epigallocatechin and catechin (**D**); and crown procyanidin consisting in a tetramer of epicatechin linked in C4-C6 and C4-C8 (**E**).

**Figure 2 molecules-25-02110-f002:**
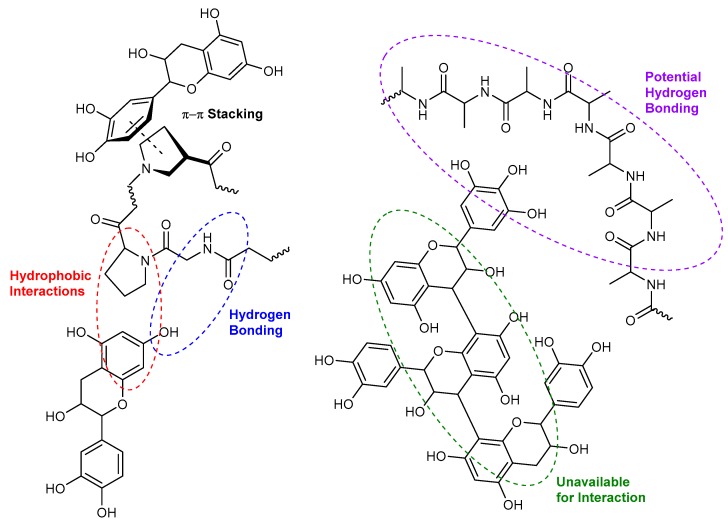
The possible hydrophobic interactions and hydrogen bondings between flavanol monomer or tannins with a protein.

**Table 1 molecules-25-02110-t001:** Tannin concentration and composition in *Vitis vinifera* grape skins and seeds, including mean degree of polymerization (mDP), terminal constitutive units (Term) and extension (Ext) units. Cat, catechin; Epi, epicatechin; EcG, Epicatechin gallate; EGC, Epigallocatechin; nd, not detected.

	*Vitis vinifera*				Term (%)			Ext (%)				
	Variety	Ripening Stage	Tannin Concentration (mg/g berry)	mDP	Cat	Epi	EcG	Cat	Epi	EcG	EGC	References
Skin	Merlot	harvest maturity		33	2.05	0.5	0.5	2.4	60.1	3.45	31	[48]
	Cabernet sauvignon	fruit-set	7.3	30	2.77	0.39	0.15	2.07	56.8	5.52	32.27	[49]
		harvest maturity	1	28	2.52	1.01	0.5	3.03	44.4	4.04	44.4	
	Shiraz	fruit-set	5.9	29	2.85	0.43	0.19	4.1	58.3	9.74	24.4	
		harvest maturity	1	31	2.58	0.41	nd	4.13	47.5	7.22	38.18	
	Albarossa	harvest maturity	11.3	13.8	5.5	1.42	0.34	20.13	58.06	4.35	10.2	[53]
	Barbera	harvest maturity	7.1	14.8	4.73	1.86	0.18	22.97	58.83	3.85	7.58	
	Nebbiolo	harvest maturity	19.2	24.2	3.47	0.62	0.04	17.08	48.15	2.83	27.79	
	Uvalino	harvest maturity	16	21.5	3.57	0.98	0.09	18.01	47.87	3.86	25.61	
	Pinot noir	at véraison	1.61	20.1								[54]
		harvest maturity	0.76	27				2.6	61.5	1.4	34.5	
Seed	Cabernet sauvignon	fruit-set	11.2	5.9	79.63	14.89	5.46	12.01	86.95	1.03	0	[51]
		harvest maturity	33.5	3.8	58.57	37.12	4.3	10.83	85.56	3.6	0	
	Syrah	after véraison	22	11	23.5	44.6	31.9	4.8	93	2.2	0	[50]
		harvest maturity	20	8	34.5	41.5	24	5.4	92.2	2.4	0	
	Albarossa	harvest maturity	53	5.2	6.08	8.1	4.40	14.74	51.56	14.51	0	[53]
	Barbera	harvest maturity	58.4	4.1	9.7	10.6	4.37	14.75	44.93	15.64	0	
	Nebbiolo	harvest maturity	73.9	4.1	10.55	8.89	4.98	11.96	50.4	13.21	0	
	Uvalino	harvest maturity	82.5	5.6	7.34	5.74	4.67	5.62	64.1	12.53	0	
	Pinot noir	at véraison	5.76	8.8							0	[54]
		harvest maturity	2.7	6.9				12.3	76.8	10.9	0	

**Table 2 molecules-25-02110-t002:** Tannin concentration in grape skin and seed of interspecific hybrid grapes, expressed as mg/g berry or mg/berry (+)-catechin equivalent.

Interspecific Hybrid Grape Variety	Total Tannin Content	Skin Tannin Content	Seed Tannin Content	References
Frontenac	0.29 mg/berry	0.03 mg/berry	0.26 mg/berry	[56]
Marquette	0.66 mg/berry	0.12 mg/berry	0.54 mg/berry	
St Croix	0.43 mg/berry	0.24 mg/berry	0.19 mg/berry	
Baco noir	0.63 mg/g berry	0.18 mg/g berry	0.45 mg/g berry	[57]
Maréchal Foch	0.96 mg/g berry	0.25 mg/g berry	0.76 mg/g berry	
Leon Millot	0.81 mg/g berry	0.22 mg/g berry	0.59 mg/g berry	

**Table 3 molecules-25-02110-t003:** Tannin concentrations measured by protein precipitation (or * methylcellulose precipitation) of wines made from *Vitis vinifera* and hybrid grape varieties.

Variety	Tannin Concentration (mg/L)	References
Pinot Noir	358	[61]
Cabernet Sauvignon	357	[57]
	2270 (epicat. equiv) *	[62]
Merlot	717	[63]
	259	[57]
Lemberger	158	
Sangiovese	174	
Cabernet franc	183	
Baco noir	49	
Maréchal Foch	83	
Corot noir	113	
Marquette	358 (epicat. equiv.) *	[62]
Noiret	354	[57]

**Table 4 molecules-25-02110-t004:** Effect of wine production on wine tannin content and mean degree of polymerization (mDP) in *Vitis vinifera* and hybrid grape varieties. ND, not determined.

Variety	Winemaking Process	Tannin Content (mg/L)	mDP	References
Pinot noir	6 days after fermentation	200	3.89	[64]
	21 days after fermentation	1000	5.89	
Cabernet sauvignon	6 days after maceration	560	11.59	[67]
	20 days after maceration	1310	13.86	
	no cold soak, 23 days after maceration	1230	12.68	
	cold soak, 23 days after maceration	1510	13.34	
Grenache	no flash détente	30.8	6.3	[68]
	flash détente, 95 °C for 6 min, pressure > 100 mbar	383	4.1	
Merlot	no microwave-assisted extraction, after 5 days crush	100	ND	[63]
	microwave-assisted extraction, after 5 days crush	210	ND	
	no microwave-assisted extraction, after 14 days crush	500	ND	
	microwave-assisted extraction, after 14 days crush	650	ND	
Maréchal Foch	must, hot press at 65 °C	152.1	3.92	[23]
	wine, hot press at 65 °C	103.9	3	
	must, 24 h cold soak	17	3.72	
	wine, 24 h cold soak	86.8	3.08	
Corot noir	must, hot press at 65 °C	158.4	5.3	
	wine, hot press at 65 °C	72	3.69	
	must, 24 h cold soak	39.7	3.72	
	wine, 24 h cold soak	145.7	3.88	
Marquette	must, hot press at 65 °C	156.3	4.66	
	wine, hot press at 65 °C	75	3.22	

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
