# Peer review of "Chemistry and Reactivity of Tannins in Vitis spp.: A Review"

_molecules, 2020, doi:10.3390/molecules25092110_

Round 1
Reviewer 1 Report
The manuscript describes the content, structure and reactivity of tanins in grapes. The main remark concerns the way the data are presented, i.e.
- better ogranisation of the manuscript, e.g. chapter 2.1. and 2.2. in one chapter; some information are several times discussed in different chapters, e.g. proteins in grapes: 5.2, 5.3 and 5.4 - it should be more concise; use more tables or scheme to present the data
- add separate chapter: biosynthesis of tannins in grape
- describe the effect of extraction or method of wine production on the content of tannins (presentation in the form of table is more readable than just text)
Author Response
We would like to thank the reviewer for the constructive comments.
Here are the responses and revisions made in the manuscript:
- Based on the reviewer comment, we reorganized slightly the manuscript. The section 2.1. and 2.2. titles have been removed to make only one chapter on “Grape species” characteristics that include the physical and chemical structure of berries. In the chapter named “Reactivity of tannins” several times proteins and cell wall material are discussed because the goal of this chapter is to show the influence of multiple factors such as the effect of tannin’s structure (5.3), other macromolecules (5.2) and grape and wine matrix (5.3) on the reactivity of tannins. The reactivity of tannins includes the interactions between tannins and proteins and cell wall material, which is the reason for discussing proteins several times in different sections.
More tables have been added to the manuscript as recommended by the reviewer. A table on the concentration of tannin in hybrid grape varieties has been added (Table 2), as well as a table of tannin concentration in wines from V. vinifera and V. spp. grapes (Table 3).
- The title “biosynthesis of tannins in grape” has been added in the chapter “Tannins in grape and wine” as requested.
- To describe the effect of winemaking process on the content of tannins, a table (Table 4) has been added in the text. This table show the tannin concentration in must and/or in wine from Vitis vinifera and interspecific hybrid grapes after different applied treatment including the time of fermentation, cold soak, extended maceration, flash détente, microwave-assisted extraction and hot press.
Reviewer 2 Report
In this manuscript, the authors have provided a detailed review on the chemistry and reactivity with other macromolecules of tannins in different grape species and wines, including both Vitis vinifera and hybrids. Systematic introduction on the grape species was included, following by the structures of different tannin groups, their profiles in different grape species and wine, and reactivity with other macromolecules such as proteins and polysaccharides. Overall, this is a very informative, well-organized review highlighting the variability of tannins in grapes and wine made from different species.
Some suggestions:
- More tables can be added. Similar to Table 1, extra tables can be added to summarize the contents of certain sections such as 4.1.2 (hybrid grapes) and 4.2 (wine).
- More discussion on the variation between grape and wine regarding to their tannin profiles can be added.
Author Response
We would like to thank the reviewer for the constructive comments.
Here are the responses and revisions made in the manuscript:
- Three tables have been added to the manuscript for an ease of understanding. A table on the concentration of tannin in hybrid grape varieties has been added (Table 2), as well as a table of tannin concentration in wines from V. vinifera and V. spp. grapes (Table 3). To describe the effect of winemaking process on the content of tannins, a table (Table 4) has been added in the text.
- A paragraph lines 340-354 has been added in the manuscript to discuss further on the variation between grape and wine tannins and Table 4 has been created in this purpose. “As previously shown by Aron and Kennedy [64], the extraction of proanthocyanidins increased significantly during alcoholic fermentation, e.g. the tannin content in Pinot noir must and wine increased from 200 to 1000 mg/L after 6 and 21 days of fermentation, respectively. Similarly, the mean degree of polymerization of tannins extracted from must after 6 days of fermentation was at 3.89 and increased to 5.89 after 21 days of fermentation. In Table 4, a summary of the effect of winemaking process on the content and size of tannins from V. vinifera and interspecific hybrid wines is shown. Skin tannins are readily extracted due to skin breaking and the presence of alcohol during fermentation and maceration. It leads to a diffusion and an extraction of tannins and anthocyanins from grape skins into the alcoholic medium, wine [39,65]. Ethanol produced by yeasts during alcoholic fermentation from grape sugars lead to a reorganization and solubilization of the grape seed lipids, which favor the extraction of tannins from seeds. This latter extraction takes more time and is slower than the extraction of tannins from skins due to the physical structure of seeds [65]. It has been previously observed that in wine the chemical structure of tannins found in wine are epigallocatechin-rich tannins extracted from grape skins rather than epicatechin gallate-rich tannins from seeds [66].”
Reviewer 3 Report
The manuscript is well-written and organized. The structure of the article provides a regular flow of information to the readers. A few minor comments:
- Lines 35-40: A reference should be included in this sentence.
- Lines 49-62: This is almost a paragraph of text without any reference. Reference(s) should be added.
- Line 109: What do the authors mean by “with potential hydrogen (pH)…”? The word “hydrogen” should be removed.
- Figure 1: The structures in Figure 1 should be re-drawn. The authors should use relative size in bond lengths. A few C-C bonds are significantly longer than others.
- Figure 2: Another type of interaction between (poly)aromatic compounds is pi-pi stacking among aromatic rings. This type of interaction is mentioned in section 5.3, where the interaction of tannins with aromatic amino-acids of proteins is described (line 567). Authors do not refer to this interaction, which is not shown in the Figure. The authors should refer to this type of interaction at this point as well. There is a lot of literature describing this type of interactions among polyphenols and it would be useful to be presented in this manuscript. Please also re-draw structures (comment as in Figure 1).
- Addition of comparative data from the literature including differences in polyphenol content prior and after fermentation, as well as in wine, would be useful.
Author Response
We would like to thank the reviewer for the constructive comments.
Here are the responses and the changes made in the manuscript based on the reviewer's comment:
- As recommended by the reviewer, two references have been added: Baxter et al., 1997 and Haslam, 1989; lines 36 and 40.
- Line 51: the following reference about the chemistry of Native American wines has been added: Rice, 1974. Line 58, a reference on the organic acids in the genus Vitis has been added: Kliewer et al., 1997.
- “potential hydrogen” did not bring further information and has therefore been removed.
- We agree with the reviewer about the different bond lengths observed in the chemical structures of Figure 1. All the structures have been re-drawn.
- Figure 2 has been adjusted and a structure of a flavanol and a proline showing the pi-pi stacking has been added. The following sentence has also been added in the text (lines 469-471) to explain this type of non-covalent interactions: “π-π stacking also exists between the aromatic rings of polyphenol and protein but has a much lower binding energy and is therefore not as common as hydrogen bonds or hydrophobic interactions (Figure 2).”
- A table (Table 4) and a paragraph has been added to the manuscript discussing the differences in tannin concentration in grapes and in wine as a result of different winemaking process.